# In Vivo Effects of Nanotechnologically Synthesized and Characterized Fluoridated Strontium Apatite Nanoparticles in the Surgical Treatment of Endodontic Bone Lesions

Faruk Oztekin [1], Turan Gurgenc [2,*], Serkan Dundar [3], Ibrahim Hanifi Ozercan [4], Mehmet Eskibaglar [1], Erhan Cahit Ozcan [5], Muhammet Bahattin Bingul [6] and Osman Habek [6]

1   Department of Endodontics, Faculty of Dentistry, Firat University, 23100 Elazig, Turkey
2   Faculty of Technology, Firat University, 23100 Elazig, Turkey
3   Department of Periodontology, Faculty of Dentistry, Firat University, 23100 Elazig, Turkey
4   Department of Pathology, Faculty of Medicine, Firat University, 23100 Elazig, Turkey
5   Department of Esthetic, Plastic and Reconstructive Surgery, Faculty of Medicine, Firat University, 23100 Elazig, Turkey
6   Department of Oral and Maxillofacial Surgery, Faculty of Dentistry, Harran University, 63300 Sanliurfa, Turkey
*   Correspondence: tgurgenc@firat.edu.tr

**Abstract:** In this study, fluoridated strontium apatite (SAP) nanoparticles with different mole percentages (5%, 10%, 30%, and 50%) synthesized using a hydrothermal method were used as biomaterials. The in vivo biocompatibility of the synthesized nanoparticles was investigated by embedding them as biomaterials in bone defects created in rat tibiae. Through the hematoxylin-eosin staining method, a histopathological analysis was performed for new bone formation, osteoblast density, and fibrotic tissue formation. Fluorine (F) addition affected the structural and morphological properties of the nanoparticles. With the F doping, the shapes of the nanoparticles changed from nano-rods to almost spherical. The Sr/P ratios, with a stoichiometric value of 1.67, were 1.76, 1.53, 1.54, 1.68, and 1.79 in pure, 5%, 10%, 30%, and 50% F-doped nanoparticles, respectively. The F/Sr ratios of 5%, 10%, 30%, and 50% F-doped nanoparticles were 0.05, 0.13, 0.16, and 0.20, respectively. The highest values in terms of fibrotic tissue formation were obtained in the group containing pure SAP. The best results in terms of new bone formation and osteoblast density in bone defects were observed in the groups with higher F ratios (30% and 50% F-doped). Pure and F-doped strontium apatite nanoparticles showed good results for new bone formation and osteoblast levels compared to the control group. It was observed that an increase in the fluorine ratio resulted in better bone healing. The results showed that pure and F-doped SAP nanoparticles synthesized by a hydrothermal method can be used as biomaterials in orthopedics and dentistry, especially in the surgical treatment of endodontic lesions.

**Keywords:** endodontic lesions; strontium apatite; fluorine; osteoblasts; new bone formation

## 1. Introduction

Apical periodontitis (AP) is an inflammatory disease that develops from the exposure of vital pulp to different oral microbiota, typically as a result of dental caries, trauma, or iatrogenic causes. The colonization of microorganisms leads to necrosis of the dental pulp and the development of infection in the periapical region of the affected teeth. As a result, activation of the host's immune response results in local acute or chronic inflammation, the resorption of periapical tissues and bone resorption, and the formation of periapical lesions [1]. Periapical problems are thought to be best treated by root canal. The success of this treatment varies between 86% and 98% [2]. When nonsurgical treatment in the maxillofacial region is not an option, periapical surgery is the only option. The reported success rates for periapical surgery range from 44–95% [3].

Endodontic surgery is very suitable for teeth that do not heal with nonsurgical endodontic treatments, and the main purpose of this treatment is to eliminate sources of infection and accelerate bone repair [4]. According to some research studies that have been conducted in recent years, retrograde filling is required to cut the connection between the root canal and the periapical tissues for successful endodontic surgical resection treatment [5,6]. For root apex filling, numerous materials have been used, such as glass ionomer, cement- and calcium-silicate-based mineral trioxide aggregate, amalgam, compomer, composite, Cavit, super ethoxy benzoic acid, and intermediate restorative material. Because of their biocompatibility, microleakage prevention, and stimulation of new bone formation, retrograde filling materials are preferred for root closure and perforation repairs [7].

Among the synthetic biomaterials, hydroxyapatite (HA) is the most widely used apatite and has Ca element content. Its formula is $Ca_{10}(PO_4)_6OH_2$, and it has chemical properties similar to bone and teeth, such as excellent bioactivity, biocompatibility, biodegradability, and nontoxicity, a noninflammatory nature, and proven osteoconductive and osteoinductive potential—these are the most important reasons why HA is preferred as a biomaterial [8–10]. Due to the properties of strontium (Sr) being similar to the element Ca, interest in the use of Sr has increased in recent years. Strontium (Sr) is an essential element for humans and is naturally prevalent in bones due to its chemical similarity to Ca [11]. Sr, one of the most important cations in hard tissues, fights bone resorption and osteoporosis by stimulating cell growth and shows high solubility, as well as osteoinduction. In addition, it has the ability to improve gene expression in osteoblastic cells and the ALP activity of mesenchymal stromal cells (MSCs), as well as inhibiting the differentiation of osteoclasts [12–17]. Previous studies have proved that the addition of Sr stimulates osteoblastic cell proliferation and differentiation and the bone regeneration of apatites [18] and that it has excellent cytocompatibility with fibroblasts [19].

Fluoride ions, as an essential trace element of human teeth and bones, are frequently preferred in apatite applications in dentistry and orthopedics [20–22]. The $F^-$ ion has good nucleation for apatites [23], and the addition of $F^-$ ions improves the lattice symmetry and structural stability of apatites [24]. F is an essential element with the capacity to improve the crystallization and mineralization of calcium phosphate in bone and dental tissue for new bone formation [25,26]. Additive F in tooth enamel improves acid resistance and creates antibacterial effects. Thus, better protection against tooth decay can be achieved [27,28]. In addition, F-doped apatites have excellent biocompatibility [29]. F is one of the most promising materials for implant surface coatings and degradable bone fragments [26,30].

The effects of F treatment depend on the concentration [31]. While it stimulates osteoblast proliferation and inhibits osteoclastic activity at low concentrations, it can have a toxic effect at very high concentrations [32]. In addition, fluoride forms fluorapatite crystals in mineralized tissues, such as bone tissue, making the tissue more resistant [33]. In addition, since the fluorine concentration in the human body is highest on the bone surface, the element fluorine is used in our study [31].

Nanoscience and nanotechnology have been frequently used in many fields, such as chemistry, physics, materials science, and biology, in recent years [34]. Apatite nanoparticles can be produced by many different methods. Solid-state reactions [35], hydrolysis [36], chemical precipitation [37], hydrothermal synthesis [38], sol-gel processes [39], and mechanochemical processes [40] are some of the most common production methods. Although each technique has its own advantages and disadvantages, the hydrothermal method has the following advantages: simplicity, low cost, homogeneity of produced particles, and good crystallinity [41,42].

Doped and undoped HAs are topics that have been extensively researched in the literature [43,44]. Studies on doped and undoped strontium apatites (SAP) are limited and remain a subject to be investigated. Although F-doped HAs have been synthesized by different researchers and their biocompatibility has been investigated, studies on the in vivo biocompatibility of F-doped apatites are still quite limited [45,46]. In addition, in our previous study, the in vivo biocompatibility of pure and Boron-doped SAP nanoparticles

was investigated [47]. In the present study, the structural and morphological properties of the pure SAP nanoparticles in our previous study are used for comparison purposes. To the best of our knowledge, F-doped SAP nanoparticles are synthesized for the first time in this study, and their in vivo biocompatibility properties are investigated. The aim of this study is to investigate the effects of fluorine-doped strontium apatite nanobiomaterials synthesized by nanotechnology on bone repair, osteoblast density, and biocompatibility in rats.

## 2. Materials and Methods

The methodology of the study is shown in Figure 1. The study was carried out under two main headings as the production of strontium apatite nanoparticles and the investigation of them in-vivo biocompatibility.

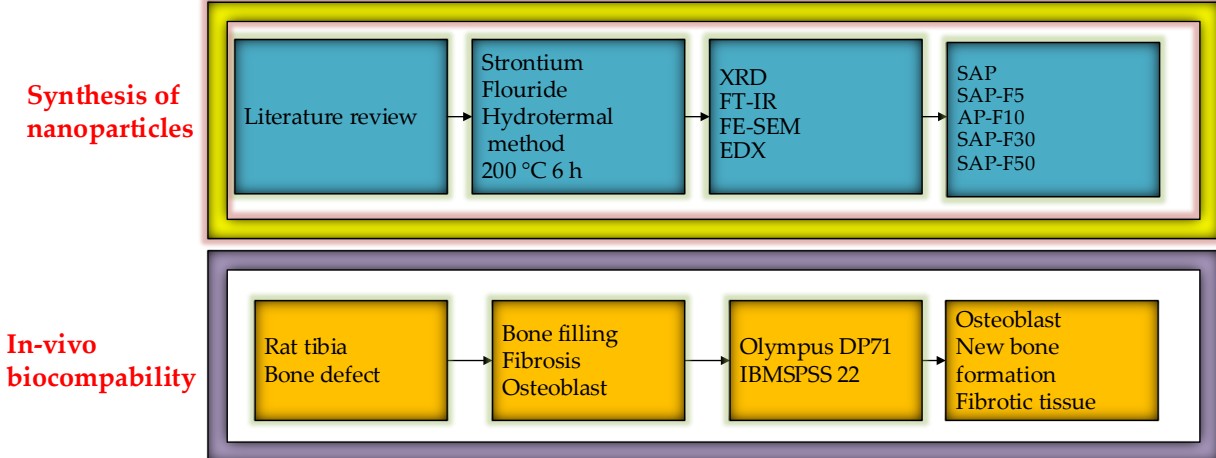

**Figure 1.** Methodology of research.

### 2.1. Synthesis and Characterization of Nanoparticles

The synthesis and characterization of pure strontium apatite nanoparticles were described in detail in a previous study by researchers in our research team [47]. F-doped strontium apatite particles were synthesized in a similar way, and the only difference was that mixtures were prepared at the rates given in Table 1 and were used in the hydrothermal method. Sigma-Aldrich (St. Louis, MO, USA) brand ammonium fluoride (NH$_4$F) was used as a fluoride source. To synthesize F-doped nanoparticles, 0.4 M strontium nitrate (Sr(NO$_3$)$_2$) was dissolved in a beaker containing 30 mL deionized water for 30 min by stirring in a magnetic stirrer (Solution A). First, 0.24 M diammonium hydrogen phosphate (H$_9$N$_2$O$_4$P) was dissolved in another beaker containing the same amount of deionized water, and then NH$_4$F was added to this solution in the ratios given in Table 1 and mixed in a magnetic stirrer for 30 min (Solution B). While stirring, Solution B was slowly added to Solution A, mixed together for 30 min, and then adjusted to pH = 10 with ammonia solution. The final solution was obtained by mixing in a magnetic stirrer and ultrasonicator for 30 and 10 min, respectively. All these procedures were performed at room temperature. The solution was poured into a Teflon-linked autoclave (Fytronix, Elazig, Turkey) and placed in a Fytronix brand hydrothermal device (Fytronix, Elazig, Turkey), and hydrothermal synthesis was performed at 200 °C for 6 h. After the device cooled down to room temperature, the precipitates were filtered with filter paper, and after the process was finished, they were cleaned several times with alcohol and deionized water, respectively. After the moist particles were dried at 80 °C for 2 h, they were ground in a mortar. The characterizations of the synthesized nanoparticles were performed with XRD (PANalytical Empyrean, Malvern, United Kingdom), FT-IR (Bruker, Billerica, MA, USA), FE-SEM (Zeiss Crossbeam 540, Oberkochen, Germany) and EDX analyses (Zeiss Crossbeam 540, Oberkochen, Germany) as detailed in our previous study [47].

**Table 1.** Additive amounts of the synthesized nanoparticles.

| Sample | Sr | P | F |
|---|---|---|---|
| | Solution A | | Solution B |
| SAP | 0.4 M | 0.24 M | - |
| SAP-F5 | 0.4 M | 0.24 M | 0.012 M |
| SAP-F10 | 0.4 M | 0.24 M | 0.024 M |
| SAP-F30 | 0.4 M | 0.24 M | 0.072 M |
| SAP-F50 | 0.4 M | 0.24 M | 0.120 M |

### 2.2. Animals and Study Design

This study was approved by the Firat University Animal Experiments Ethics Committee with the decision of 2021/16. All recommendations of the World Medical Association Declaration of Helsinki were studied for the protection of the laboratory test animals. Using a statistical power analysis, it was calculated that there should be at least eight rats in each group. In our study, considering the risk of death of the animals during or after the surgical procedure, a total of 66 rats, 11 rats in each group, were used. Bone defects of 2.5 mm in diameter and 4 mm in depth were formed in the right tibia of the rats, which were randomly divided into six groups ($n = 11$). No procedure was performed on the control group, except for creating a defect in Group 1 ($n = 11$). Pure and fluorine-doped SAP nanoparticles, which were produced nanotechnologically by a hydrothermal method, were placed in experimental groups as follows: Group 2 strontium ($n = 11$), Group 3 strontium with 5% fluorine ($n = 11$), Group 4 strontium with 10% fluorine ($n = 11$), Group 5 strontium with 30% fluorine ($n = 11$), and Group 6 strontium with 50% fluorine ($n = 11$).

### 2.3. Surgical Procedure

Surgical procedures were performed under general anesthesia. After full-thickness dissection, the metaphyseal part was reached by making bone contact with the scalpel over the tibial crest. Bone defects with a height of 4 mm and a diameter of 2.5 mm were created under physiological saline cooling. After the surgical procedure, it was closed with 30 dissolving sutures. Postoperative antibiotics (penicillin 50 mg/kg) and pain relievers (tramadol hydrochloride 0.1 mg/kg) were administered for infection prevention and pain control. At the end of the study period, all the rats were euthanized. Biomaterials placed on the right tibia bones were decalcified, and histological bone filling, fibrosis, and osteoblast analyses were performed. Due to rat deaths in the groups during the experimental protocol, the study was continued with eight rats in each group.

### 2.4. Histological Procedure

Tibias were kept in 10% formaldehyde for 72 h and then demineralized in 10% formic acid. After demineralization, all the tibia bones were dried, embedded in paraffin, and made ready for histological section. Microscopic analysis was performed by hematoxylin-eosin staining. The prepared samples were examined in 6 μm thick sections under a light microscope. Cells with no osteoblasts were scored = 0, cells with mild osteoblasts = 1, cells with moderate osteoblast = 2, and cells with dense osteoblasts = 3. No new bone formation was scored as = 0, low new bone formation = 1, moderate new bone formation = 2, and extensive new bone formation = 3. No visible fibrotic tissue was scored = 0, superficial or focal fibrotic tissue appearance = 1, superficial diffuse or deep localized fibrotic tissue = 2, and deep and extensive fibrotic tissue formation = 3 [48].

### 2.5. Statistical Analysis

Statistical analyses of the data were performed with the IBM SPSS 22 statistical program (Endicott, New York, ABD). To determine whether the data showed a normal distribution, the Shapiro–Wilk test was used. The data that did not show normal distribution were presented as the median (Min–Max). The Kruskal–Wallis H test was used to compare more

than two independent groups for non-normally distributed quantitative data, followed by the Dunn–Bonferroni post hoc test for pairwise comparisons. The statistical significance level was determined to be $\alpha = 0.05$.

## 3. Results

The X-ray diffraction (XRD) analysis results of the pure and F-doped SAP nanoparticles are shown in Figure 2. The XRD results of the synthesized nanoparticles were compatible with JCPDS card no. 33-1348. The peaks seen in the pure SAP sample at 20.99°, 24.46°, 26.66°, 27.90°, 30.58°, 31.70°, 38.35°, 41.55°, 44.65°, 46.04°, 46.96°, 48.51°, and 49.95° 2θ angles were characteristic peaks of SAP nanoparticles. The same characteristic peaks were also found in the F-doped samples. The intensities and positions of the peaks were affected by the F contribution. As the F contribution increased, the intensity of the peaks increased. No peaks belonging to a secondary phase or impurity were found in any of the synthesized samples.

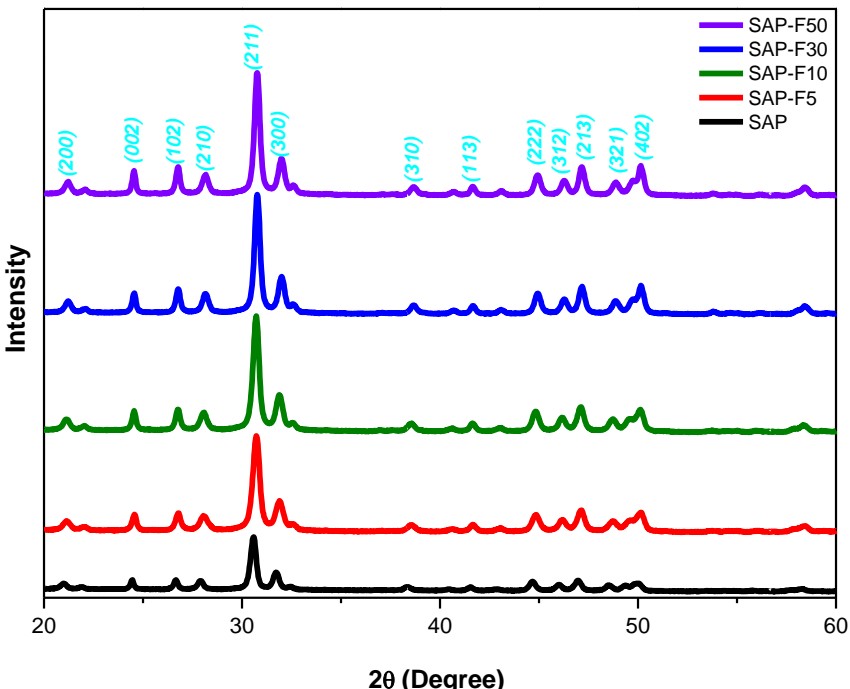

**Figure 2.** XRD analysis results of synthesized nanoparticles.

The FT-IR analysis results of the pure and F-doped SAP nanoparticles are shown in Figure 3. Changes in peak intensities and positions were observed with the F doping. In the FT-IR analysis, peaks of $PO4^{3-}$, $OH^-$, $CO_3^{2-}$, and $HPO_4^{2-}$ were detected.

Figure 4 shows the field emission scanning electron microscope (FE-SEM) images of the pure and F-doped SAP nanoparticles. As can be seen, the synthesized particles were composed of nanostructures. The pure sample consisted of nano-rod-shaped structures. With the F doping, the structures changed from nano-rod to a nearly spherical shape and agglomerated. The EDX analysis graphs of the synthesized nanoparticles are shown in Figure 5, and the EDX analysis results are given in Table 2. The nanoparticles were composed of S, P, F, and O elements. As expected, the F ratio in the SAP nanoparticles increased as the F-doping ratio increased. For the pure, 5%, 10%, 30%, and 50% F-doped samples, the Sr/P ratios were 1.76, 1.53, 1.54, 1.68, and 1.79, respectively. As the F-doping ratio increased, the P ratio decreased, and the Sr/P ratio increased. The F/Sr ratios ranged from 0.05 to 0.2 and increased as the F-doping ratio increased.

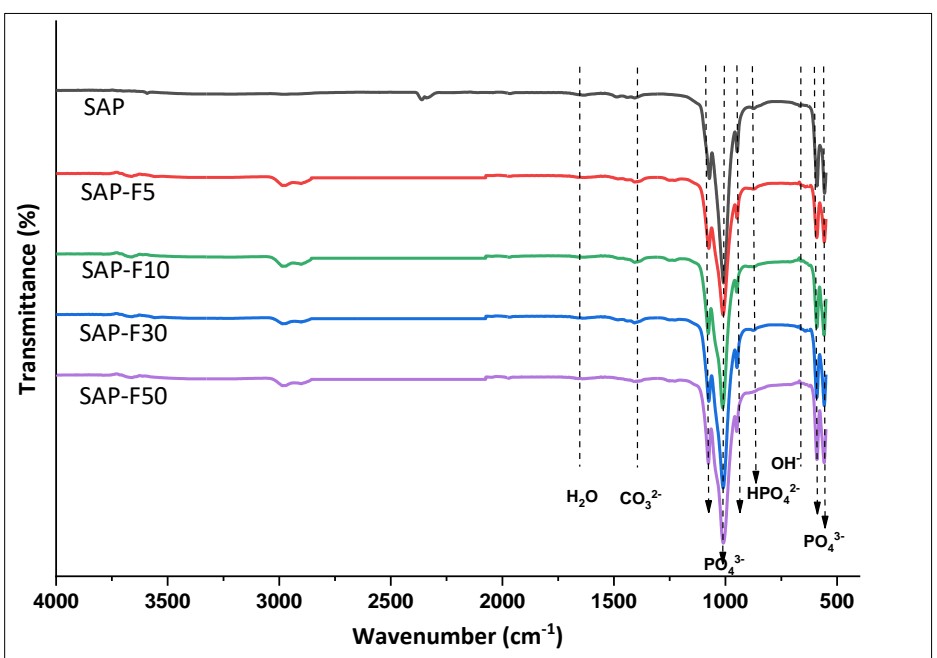

**Figure 3.** FT-IR analysis results of synthesized nanoparticles.

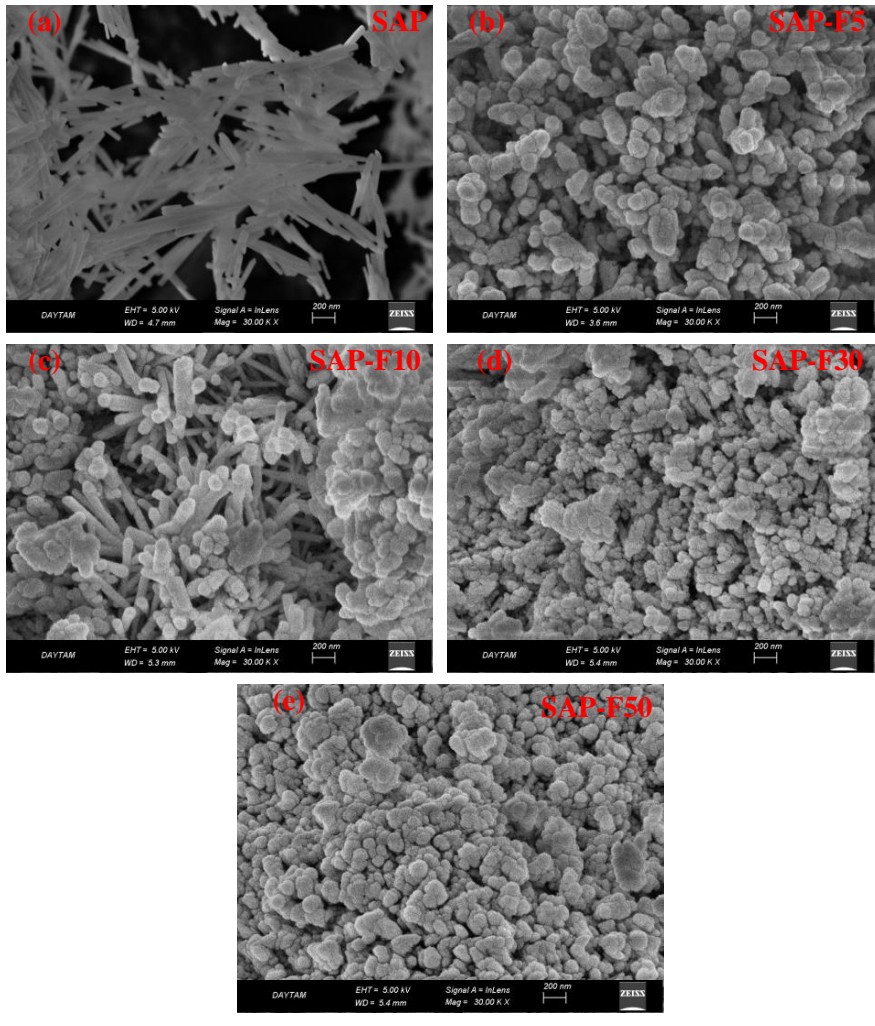

**Figure 4.** FE-SEM images of synthesized nanoparticles.

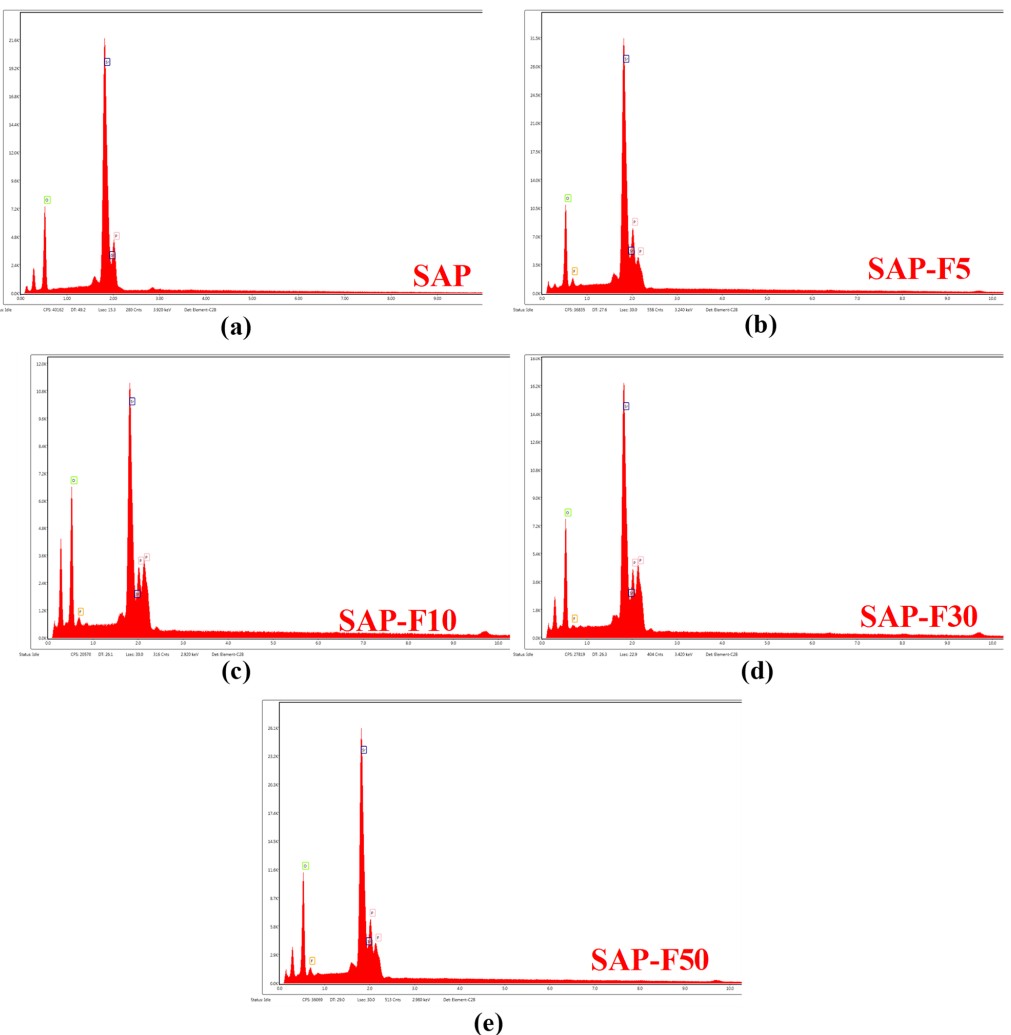

**Figure 5.** EDX analysis graphics of synthesized nanoparticle.

**Table 2.** EDX analysis results of fabricated nanoparticles (at.%).

| Sample | Other Elements | Sr | P | F | Sr/P | F/Sr |
|---|---|---|---|---|---|---|
| SAP | 60.49 | 25.19 | 14.32 | - | 1.76 | - |
| SAP-F5 | 57.39 | 24.98 | 16.33 | 1.30 | 1.53 | 0.05 |
| SAP-F10 | 58.18 | 23.57 | 15.27 | 2.98 | 1.54 | 0.13 |
| SAP-F30 | 56.17 | 24.93 | 14.83 | 4.07 | 1.68 | 0.16 |
| SAP-F50 | 53.54 | 26.41 | 14.77 | 5.28 | 1.79 | 0.20 |

According to the statistical results of our study, the best results in terms of osteoblast level and new bone formation were seen in the groups with higher added fluorine. There was a statistically significant difference only between the control group and the strontium with 30% fluorine and strontium with 50% fluorine groups. There was a difference between the other groups in terms of these parameters, but it was not statistically significant. Similar results were obtained in the groups in terms of fibrotic tissue care. However, there was a statistically significant difference between only the strontium group, the control group, and the strontium with 10% fluorine group. According to the results of the study, it was observed that the osteoblast level and new bone formation were higher in the groups with high fluoride content. In terms of the fibrotic tissue parameter, although all the groups gave similar results, it was observed that it was more advanced in the strontium group

without fluorine supplementation (Table 3). Boxplot graphs of the obtained statistical results are shown in Figure 6. The long and short boxes in the graphs show the distribution of the data. The longer the boxes, the more scattered the data. The median line shows the midpoint of the obtained data. If the median line of one boxplot is outside the box of another boxplot, a difference is likely between the two groups. When the median lines of the groups were compared with each other in the graphs, it was seen that the statistical results were supported, as given in Table 3.

**Table 3.** Intergroup comparison of variables.

| | Osteoblast Median (Min–Max) ($n$ = 8) | New Bone Formation Median (Min–Max) ($n$ = 8) | Fibrotic Tissue Median (Min–Max) ($n$ = 8) |
|---|---|---|---|
| **Control (1)** | 1 (1–2) | 1 (1–2) | 0 (0–0) |
| **Strontium (2)** | 2 (2–2) | 2 (1–2) | 2 (1–2) |
| **Strontium with 5% fluorine (3)** | 2 (1–2) | 2 (1–2) | 0.5 (0–1) |
| **Strontium with 10% fluorine (4)** | 2 (1–2) | 2 (1–2) | 0 (0–1) |
| **Strontium with 30% fluorine (5)** | 2.5 (2–3) | 2.5 (2–3) | 0 (0–2) |
| **Strontium with 50% fluorine (6)** | 2.5 (2–3) | 2.5 (2–3) | 1 (0–1) |
| ***p*** | <0.001 | <0.001 | <0.001 |
| ***p*** * | 1–5: $p$ = 0.002 <br> 1–6: $p$ = 0.002 <br> Others: $p$ > 0.05 | 1–5: $p$ = 0.001 <br> 1–6: $p$ = 0.001 <br> Others: $p$ > 0.05 | 1–2: $p$ < 0.001 <br> 2–4: $p$ = 0.004 <br> Others: $p$ > 0.05 |

$p$: Kruskal–Wallis H test; $p$ *: Dunn–Bonferroni test (pairwise).

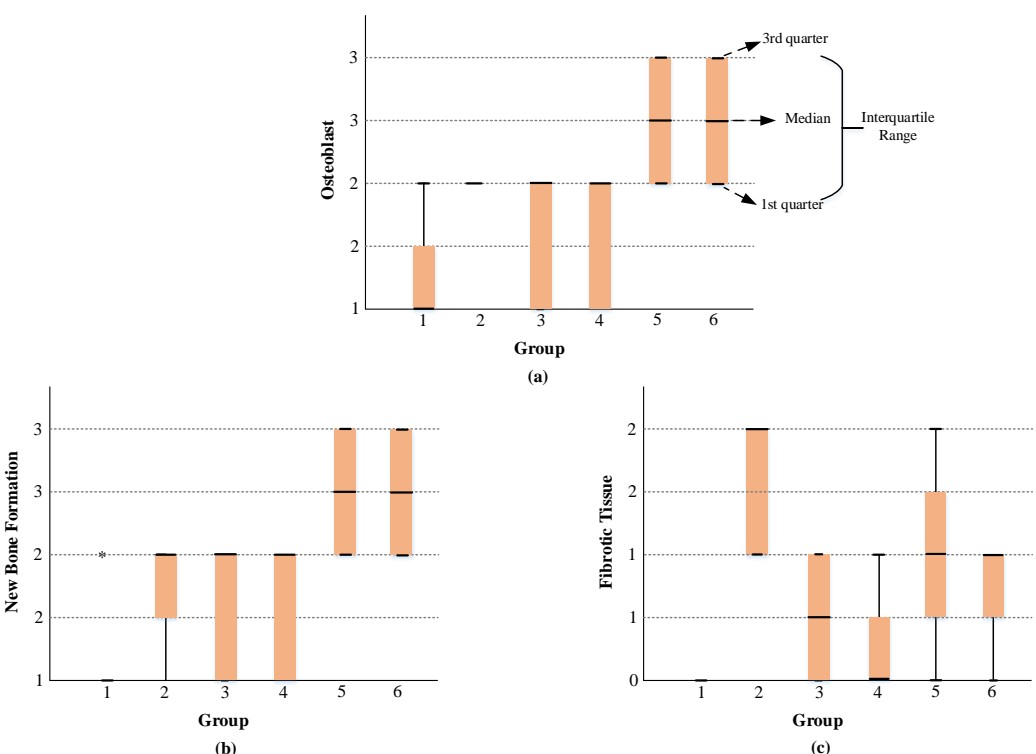

**Figure 6.** Statistical results of (**a**) osteoblast, (**b**) new bone formation, and (**c**) fibrotic tissue. (* = Extreme outlier value).

In our histological analysis, changes were seen in the bone defects created in the rat tibia and placed with the biomaterial. The density of the osteoblast cells in the groups, the new bone formation, and the fibrotic tissue formation are shown in Figure 7. Our findings revealed that doping strontium apatite with fluorine made the material more biocompatible and increased new bone formation.

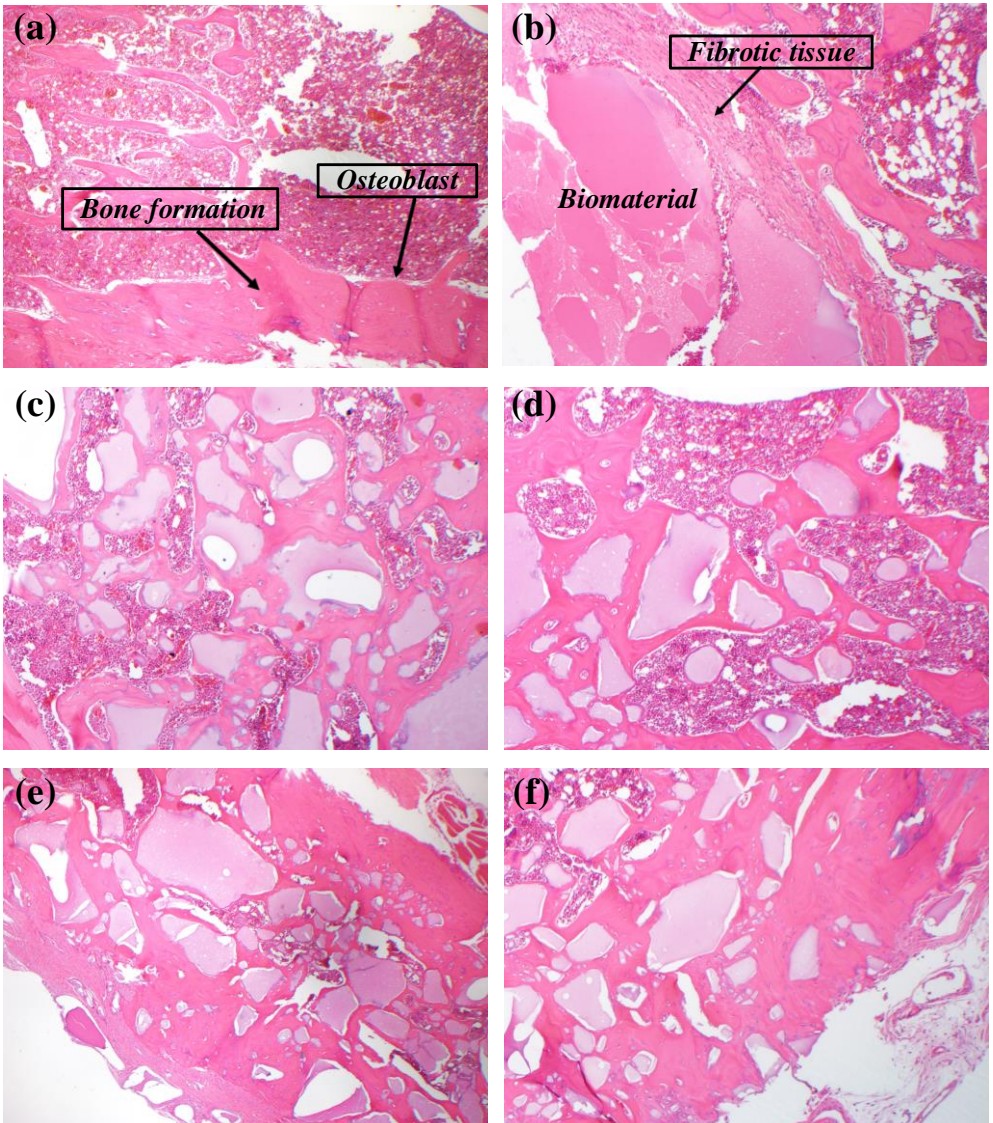

**Figure 7.** Histopathological images of groups (**a**) Control, (**b**) SAP, (**c**) SAP-F5, (**d**) SAP-F10, (**e**) SAP-F30, and (**f**) SAP-F0.

## 4. Discussion

The characteristic XRD peaks of pure SAP nanoparticles corresponded to the (200), (002), (102), (210), (211), (300), (310), (113), (222), (312), (213), (321), and (402) planes [27,41]. It was observed from the XRD analysis that the peak values in the (210) plane increased as the fluorine contribution increased. The increase in intensity with the increase in fluorine additives indicated increased crystallinity [49]. In studies on F-doped hydroxyapatite nanoparticles in the literature, it has also been observed that high-density peaks shift to higher 2θ angles with F doping [45,50]. It is thought that the changes in the intensities and positions of the XRD diffraction peaks with the F contribution are due to the fact that the ionic radii of $F^-$ (132 p.m.) is lower than that of $OH^-$ (168 p.m) [46,51]. In this case, by increasing the crystallinity and stability of the nanoparticles, it may have led to an increase in the intensity of the peaks and a shift of their positions to higher 2θ values [51,52]. The XRD analysis results are in agreement with the literature [45,53–55].

In the FT-IR analysis results, the peaks between 552 cm$^{-1}$ and 593 cm$^{-1}$, as well as at 923 cm$^{-1}$, are ascribed to the bending vibration of the $PO_4^{3-}$ functional group [56,57]. The small absorption band observed at about 700 cm$^{-1}$ is attributed to $OH^-$ ions [58]. The peak observed at about 856 cm$^{-1}$ is ascribed to $HPO_4^{2-}$ [59,60]. The absorption

band seen at about 994–1060 cm$^{-1}$ is ascribed to the stretching vibration of PO$_4^{3-}$ [53,55]. The peak at about 1452 cm$^{-1}$ corresponds to CO$_3^{2-}$ ions [55,59]. The absorption band observed at 1574 cm$^{-1}$ is ascribed to the bending mode of H$_2$O molecules. The band at about 3510 cm$^{-1}$ corresponds to the stretching mode of H$_2$O molecules [58]. The peaks observed between about 2750–3000 cm$^{-1}$ and about 3640 cm$^{-1}$ in the F-doped samples are attributed to the OH–F bond [61,62]. It is thought that the changes in the positions and intensities of the absorption peaks are caused by the changes in structure, conformation, and intermolecular interaction due to the change in the F-doping ratio [52,63]. These peaks indicate that the OH$^-$ and F$^-$ ions were replaced in the apatite structure, and F doping was achieved successfully.

Doping with F changed the morphology of nanoparticles and increased agglomeration. In addition, the F doping reduced the size of the nanoparticles. Due to the low ionic radius of F compared to OH, the nanoparticles decreased in size, and it is thought that the nanoparticles tend to agglomerate with increasing crystallinity and surface area [50,63]. It has also been stated in studies that have been carried out by other researchers that F doping increases agglomeration and causes a decrease in particle sizes [45,50,64]. The presence of Sr, P, and F elements in the EDX analyses confirms the successful synthesis of the F-doped strontium apatites [54,65]. The Sr/P ratios of the samples were close to the stoichiometric Sr/P ratio (1.67), and the sample closest to this ratio was the SAP-F30 sample (1.68). It is thought that the increase in the Sr/P ratio as the F-doping ratio increases was due to the smaller size of the F$^-$ ions compared to the OH$^-$ ions. This may be due to the fact that, at high F concentrations, F$^-$ ions with lower ionic radii are incorporated into the structure of SAP nanoparticles faster than OH$^-$ ions [49]. The F/Sr ratio is limited to 0.2 in apatites, and this ratio was not exceeded in our study [49,66]. The structural and morphological analyses showed that pure and F-doped SAP nanoparticles were successfully synthesized by the hydrothermal method.

Bone defects have been repaired using a variety of techniques and agents to date [67]. One of the agents used in these treatment processes is strontium. Strontium provides mineralization without damaging the bone tissue and stimulates osteoblastic activity by decreasing osteoclastic activity. In addition, since strontium has been a frequently used biomaterial in bone tissue research in recent years, we used it as a repair material in bone defects in our study [68]. A group of researchers investigated the effect of strontium on mouse calvarial cells and showed that the presence of strontium increased osteoblast density [69]. In another study, it was determined that hydroxyapatite placed on a bilaterally defected tibia and observed for 20, 30, or 45 days was biocompatible in direct bone contact during the bone neoformation process, but there was no acceleration in bone healing compared to the control [70]. In our previous study, we also concluded that new bone formation was faster when strontium apatite was used instead of hydroxyapatite material with similar properties [47].

Fluorine is one of the elements necessary for the development of mineralized tissues, such as bone. Fluorine stimulates the differentiation of osteoblasts and inhibits the activity of osteoclasts, allowing for the growth of bone tissue [71]. In addition, while fluorine stimulates the proliferation of bone cells at the cellular level by directly inhibiting phosphotyrosyl protein phosphatase activity, it increases the levels of the insulin growth factor-1 and transforming growth factor-1 (TGF-β1) mediators and ensures tissue growth and repair [31,72]. Due to their positive properties in bone tissue, nanoparticles containing different fluorine concentrations were used in our study. The effects of pure and fluorine-doped strontium apatite nanoparticles on bone defects have not been examined in the literature. For this reason, pure and fluorine-doped strontium apatite nanoparticles were used at different concentrations in our study. The effects of these biomaterials on new bone formation, osteoblast density, and fibrotic tissue formation in bone defects were evaluated. While it was observed that 50% fluorine-doped SAP nanoparticles provided better results in terms of osteoblast density and new bone formation, intensive fibrotic tissue formation was observed in the pure-SAP-implanted group. The ionic form of F stimulated the pro-

liferation of bone cells by directly inhibiting phosphotyrosyl protein phosphatase activity. This increased the total cellular tyrosyl phosphorylation, thereby leading to the stimulation of bone cell proliferation. In our study, it was thought that the new bone formation and high osteoblast density were due to the stimulation of the pathways involved in osteogenic activity by the ionic form of F [31].

## 5. Conclusions

Fluoridated strontium apatite nanoparticles were successfully synthesized using a hydrothermal method. Based on the study's limited findings, local fluoride and strontium application may be materials that could be used successfully as retrograde fillings during apical surgery. Furthermore, it was possible that local fluoride application may be more beneficial in terms of bone healing. Pure or fluorine-doped strontium apatite nanoparticles could be useful as a biomaterial in dental and orthopedic applications. The use of pure or fluorine-doped strontium apatite nanoparticles as biomaterials in dental and orthopedic applications may be advantageous. However, it is suggested that fluorine may be harmful, and studies should be conducted to evaluate the concentrations.

**Author Contributions:** F.O., T.G. and S.D.: conceived the ideas. S.D.: animal surgical procedure. F.O., T.G. and M.E.: led the writing. F.O., T.G., S.D., I.H.O., M.E. and E.C.O.: collected and analyzed the data. F.O., T.G., S.D., I.H.O., M.E., E.C.O., M.B.B. and O.H.: designed the manuscript; worked with graphic material; and edited and processed the manuscript. All authors have read and agreed to the published version of the manuscript.

**Funding:** This research received no external funding.

**Institutional Review Board Statement:** This study was evaluated by the Firat University Animal Experiments Ethics Committee, and the study design was approved by decision 2021/16. All recommendations of the World Medical Association Declaration of Helsinki for the protection of laboratory test animals were followed. We confirm that all methods were carried out in accordance with relevant guidelines and regulations.

**Informed Consent Statement:** Not applicable.

**Data Availability Statement:** The data presented in this study are available on request from the corresponding author.

**Conflicts of Interest:** There is no conflict of interest in the study.

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
