# Peer review of "In Vivo Effects of Nanotechnologically Synthesized and Characterized Fluoridated Strontium Apatite Nanoparticles in the Surgical Treatment of Endodontic Bone Lesions"

_crystals, doi:10.3390/cryst12091192_

Round 1
Reviewer 1 Report
Authors in their manuscript explained the Characterization and In-Vivo Properties of Nanotechnologically Synthesized Fluoridated SrP Nanoparticles. The manuscript needs some major revision:
- In the material and method section, the authors have used a flow chart to describe the methodology. Generally, a flow chart is used to explain the method in ppts or reports. In a research paper, the method is explained in a text and after that, the schematic can be made for a better understanding of the reader or to make the paper interesting.
- In the Result section part, an explanation of graphs should be in detail. Like in the explanation of XRD and FTIR Spectrum, the authors wrote that the positions of the peaks were shifted and the intensities were changed with the increment in the F concentration. So the reason behind the change in peak intensity and shifting should be given.
- In the FE-SEM analysis, the authors observed that the morphology of strontium apatite nanoparticles was tuned from rod to spherical and then to aggregates with the doping of fluorine. The reason behind the altering of morphology should be mentioned.
- Table 2 represents the at % of Sr, P, O, and F. The table should be edited and not include the at % of O. As you know oxygen is present in the atmosphere, so the EDX will capture and show you the at %. It is a general thing, you can take ideas from the literature about the interpretation of the EDX spectrum.
- In the discussion session, the author wrote that the effect of pure and fluorine-doped SAP nanoparticles on bone defects has not been investigated in the literature. But in line number 211 at the beginning of the discussion section, the authors give references {{44, 48)} which are based on similar studies. The authors need to go through the literature carefully.
- The authors must follow one style to write the references. Either they should use the full name or abbreviation of the journal but not both while writing. In some places, the dot is added after the journal name and in some places, it is missing. Therefore, there are certain things to keep in mind while writing the reference
Finally, the authors need to pay more attention to the result explanation part which is a major flaw in the paper
Author Response
Respond to Reviewer
We sincerely thank the reviewer for constructive criticisms and valuable comments, which were of great help in revising the manuscript. The comments, which significantly contributed to improving the quality of the publication are quite helpful for us. Accordingly, the revised manuscript has been systematically improved with new information and additional interpretations. Please find below a detailed response to the each of the comments. And we hope the Editors and the Reviewer will be satisfied with our responses to the ‘comments’ and the revisions for the original manuscript.
Reviewer #1:
Authors in their manuscript explained the Characterization and In-Vivo Properties of Nanotechnologically Synthesized Fluoridated SrP Nanoparticles. The manuscript needs some major revision:
- In the material and method section, the authors have used a flow chart to describe the methodology. Generally, a flow chart is used to explain the method in ppts or reports. In a research paper, the method is explained in a text and after that, the schematic can be made for a better understanding of the reader or to make the paper interesting.
- In the Result section part, an explanation of graphs should be in detail. Like in the explanation of XRD and FTIR Spectrum, the authors wrote that the positions of the peaks were shifted and the intensities were changed with the increment in the F concentration. So the reason behind the change in peak intensity and shifting should be given.
- In the FE-SEM analysis, the authors observed that the morphology of strontium apatite nanoparticles was tuned from rod to spherical and then to aggregates with the doping of fluorine. The reason behind the altering of morphology should be mentioned.
- Table 2 represents the at % of Sr, P, O, and F. The table should be edited and not include the at % of O. As you know oxygen is present in the atmosphere, so the EDX will capture and show you the at %. It is a general thing, you can take ideas from the literature about the interpretation of the EDX spectrum.
- In the discussion session, the author wrote that the effect of pure and fluorine-doped SAP nanoparticles on bone defects has not been investigated in the literature. But in line number 211 at the beginning of the discussion section, the authors give references {{44, 48)} which are based on similar studies. The authors need to go through the literature carefully.
- The authors must follow one style to write the references. Either they should use the full name or abbreviation of the journal but not both while writing. In some places, the dot is added after the journal name and in some places, it is missing. Therefore, there are certain things to keep in mind while writing the reference
Finally, the authors need to pay more attention to the result explanation part which is a major flaw in the paper
To Reviewer 1:
- The authors would like to thank the reviewer for his/her suggestion. The figure showing the methodology of the paper was removed and the methodology was explained in the text in detail. The figure containing the methodology of the study was given under the material method section.
- Thank you very much for your suggestions to make the study stronger. Added explanations of graphics that are not explained in the results section. In the discussion part, information was given about the causes of the changes in the positions and intensities of the XRD and FT-IR peaks due to the F contribution.
- The authors would like to thank the reviewer for his/her valuable comments on this subject. Necessary explanations about the reason for the changes in the morphology of the nanoparticles due to the F doping were added to the relevant section.
- The authors would like to thank the reviewer for his/her suggestion. O was removed from the table containing the EDX results and the text was rearranged accordingly.
- The authors would like to thank the reviewer for his/her suggestion on how to make our paper stronger. The part you specified in the introduction has been corrected as follows.
‘’Although F doped HAs have been synthesized by different researchers and their bio-compatibility has been investigated, studies on the in vivo biocompatibility of F doped apatites are still quite limited [45,46]. Also in our previous study, the in vivo biocom-patibility of pure and Boron-doped SAP nanoparticles were investigated [47]. In the present study, the structural and morphological properties of pure SAP nanoparticles in our previous study were used for comparison purposes. To the best of our knowledge, F-doped SAP nanoparticles were synthesized for the first time in this study, and their in vivo biocompatibility properties have been investigated. The aim of this study was to investigate the effects of fluorine-doped strontium apatite nanobi-omaterials synthesized by nanotechnology and on bone repair, osteoblast density, and biocompatibility in rats.’’
The part you mentioned in the discussion section has been rearranged as follows;
‘’In studies on F-doped hydroxyapatite nanoparticles in the literature, also it has been observed that high-density peaks shift to higher 2θ angles with F-doping [45,50].’’
- Thank you for taking the time to spot this issue. References have been rearranged according to article format.

Reviewer 2 Report
"Characterization and In-Vivo Properties of Nanotechnologi- 2 cally Synthesized Fluoridated SrP Nanoparticles" by ÖZTEKİN et al is a well thought and conceptualized paper with a novel idea. With some minor modifications it will be suitable for publication.
Abstract : Although it is well written, it seems to be lack in attracting the readers. May be add some numerical parameters that will make it more attractive and engage the reader.
Introduction: The following recent paper was about fluorine substituted HA for related endodontic application. It would be better if you include this kind of related reference and discuss at the relevant place. Acta Biomaterialia 79 (2018) 148-157.
Line 56 to line 59 : rewrite the lines they seem monotonous. May be condense the matter.
Figure 1 : Try to change the flowchart it looks confusing. May be go for a vertical flowchart
Line 144 : The scoring is confusing and it would have been better if instead of scoring explain based on the microscopic analysis. or make this scoring pattern more clearer manner.
Figure 6 : Please add a clear image as the image is blurred.
Author Response
Respond to Reviewer
We sincerely thank the reviewer for constructive criticisms and valuable comments, which were of great help in revising the manuscript. The comments, which significantly contributed to improving the quality of the publication are quite helpful for us. Accordingly, the revised manuscript has been systematically improved with new information and additional interpretations. Please find below a detailed response to the each of the comments. And we hope the Editors and the Reviewer will be satisfied with our responses to the ‘comments’ and the revisions for the original manuscript.
Reviewer #2:
"Characterization and In-Vivo Properties of Nanotechnologi- 2 cally Synthesized Fluoridated SrP Nanoparticles" by ÖZTEKİN et al is a well thought and conceptualized paper with a novel idea. With some minor modifications it will be suitable for publication.
- Abstract : Although it is well written, it seems to be lack in attracting the readers. May be add some numerical parameters that will make it more attractive and engage the reader.
- Introduction: The following recent paper was about fluorine substituted HA for related endodontic application. It would be better if you include this kind of related reference and discuss at the relevant place. Acta Biomaterialia 79 (2018) 148-157.
- Line 56 to line 59 : rewrite the lines they seem monotonous. May be condense the matter.
- Figure 1 : Try to change the flowchart it looks confusing. May be go for a vertical flowchart
- Line 144 : The scoring is confusing and it would have been better if instead of scoring explain based on the microscopic analysis. or make this scoring pattern more clearer manner.
- Figure 6 : Please add a clear image as the image is blurred.
To Reviewer 2:
- The authors would like to thank the reviewer for his/her valuable comments on this subject. The following part has been added to the abstract.
‘’Fluorine (F) addition affected the structural and morphological properties of nanoparticles. As the F doping, the shapes of the nanoparticles changed from nano-rod to almost spherical. The Sr/P ra-tio with a stoichiometric value of 1.67 is 1.76, 1.53, 1.54, 1.68 and 1.79 in pure, 5% F, 10% F, 30% F and 50% F doped nanoparticles, respectively. The F/Sr ratios of 5% F, 10% F, 30% F and 50% F doped nanoparticles are 0.05, 0.13, 0.16 and 0.20, respectively.’’
- The authors would like to thank the reviewer for his/her suggestion. The relevant work is used where necessary.
- The authors would like to thank the reviewer for his/her recommendations. The mentioned part has been rearranged as below;
‘’Sr, one of the most important cations in hard tissues, fights bone resorption and osteo-porosis by stimulating cell growth, and shows high solubility as well as osteoinduction. Besides, it has the ability to improve gene expression in osteoblastic cells and ALP ac-tivity of mesenchymal stromal cells (MSCs) as well as inhibit differentiation of osteo-clasts.’’
- Thank you for taking the time to spot this issue. Flowchart has been removed and this part is explained as text. Added a figure giving the methodology of the study.
- Thank you very much for your suggestions to make the study stronger. Scoring is explained more clearly and added to the relevant section.
- Thank you for taking the time to spot this issue. Figure 6 has been rearranged.

Round 2
Reviewer 1 Report
The authors have addressed all comments and explained well in the manuscript. Only one comment needs to be re-corrected which is given below:- Remove the oxygen atom from the line number. 281. As mentioned in the previous comments, oxygen comes from the environment, so its presence will appear.